# Paraaortic Lymphadenectomy in Gynecologic Oncology—Significance of Vessels Variations

**DOI:** 10.3390/jcm11040953

**Published:** 2022-02-11

**Authors:** Stoyan Kostov, Ilker Selçuk, Angel Yordanov, Yavor Kornovski, Hakan Yalçın, Stanislav Slavchev, Yonka Ivanova, Svetla Dineva, Deyan Dzhenkov, Rafał Watrowski

**Affiliations:** 1Department of Gynecology, University Hospital “Saint Anna”, Medical University of Varna, 9002 Varna, Bulgaria; drstoqn.kostov@gmail.com (S.K.); ykornovski@abv.bg (Y.K.); st_slavchev@abv.bg (S.S.); yonka.ivanova@abv.bg (Y.I.); 2Department of Gynecologic Oncology, Ankara City Hospital, Woman’s Health Training and Research Hospital, University of Health Sciences, Ankara 06230, Turkey; ilkerselcukmd@hotmail.com (I.S.); drhyalcin@yahoo.com (H.Y.); 3Department of Gynecologic Oncology, Medical University Pleven, 5800 Pleven, Bulgaria; 4Diagnostic Imaging Department, Medical University of Sofia, 1431 Sofia, Bulgaria; svetladineva7@gmail.com; 5National Cardiology Hospital, 1309 Sofia, Bulgaria; 6Department of General and Clinical Pathology, Forensic Medicine and Deontology, Faculty of Medicine, Medical University—Varna “Prof. Dr. Paraskev Stoyanov”, 9002 Varna, Bulgaria; ddzhenkov@mail.bg; 7Faculty of Medicine, University of Freiburg, 79106 Freiburg, Germany; rafal.watrowski@gmx.at

**Keywords:** paraaortic lymph node dissection, vessels variations, surgical classification

## Abstract

Lymphadenectomy has been an essential part of the surgical treatment in surgical oncology, as the lymphatic channels and nodes are the main dissemination pathway for most of the gynecological cancers. Pelvic and paraaortic lymphadenectomy are frequent surgical procedures in gynecologic oncology. Paraaortic lymph node dissection facilitates staging, prognosis, surgical and postoperative management of patients. It is one of the most challenging retroperitoneal surgeries. A comprehensive knowledge of the paraaortic region is mandatory. Intraoperative bleeding is the most common complication during lymphadenectomy due to direct vascular injury, poor tissue handling, exuberant retraction and possible anatomical variations of the vessels in the paraaortic region. Approximately, one-third of women will have at least one anatomic variation in the paraaortic region. It must be stressed that anomalous vessels may be encountered in every woman who will undergo surgery. Consequently, detailed knowledge of anatomical vessels variations is required in order to prevent iatrogenic vessel injury. The importance of these variations is well described in urology, vascular and general surgery. Conversely, in oncogynecological surgery, there are few articles, which described some of the vessels variations in the paraaortic region. The present article aims to propose a surgical classification and to describe the majority of vessels variation, which could be encountered during paraaortic lymphadenectomy in gynecologic oncology. Moreover, surgical considerations in order to prevent anomalous vessels injury are well described.

## 1. Introduction

Paraaortic lymph node dissection (PALND) facilitates staging, prognosis, surgical and postoperative management of the patients [1,2]. PALND is part of the surgical treatment of the majority of gynecological malignancies [2]. The majority of articles that have been extensively cited in the gynecological literature discussed the diagnostic and therapeutic role of PALND, as there are many uncertainties [3,4]. Therefore, there are few medical articles describing the retroperitoneal paraaortic lymph nodes (PALNs) anatomy, the extent and the anatomical boundaries of dissection, which are required during PALND [5]. Moreover, the significance of anatomical variations during PALND has been recently stated in a few original articles and case reports. There is no study or classification summarizing most of the vessels variations in the paraaortic region. Oncogynecologists should be well acquainted with retroperitoneal anatomy and vessels variations for achieving complete and safe PALND. Learning the proper lymphadenectomy technique and awareness of possible anatomical variations may prevent iatrogenic vessel injury [1]. The goal of the present article is to elaborate and emphasize the PALNs’ anatomy, definition, anatomical landmarks and to highlight the wide range of vascular variations during PALND.

## 2. Topographical Anatomy in the Paraaortic Regions and Paraaortic (Lumbar) Lymph Nodes’ Anatomical Definition

Topographical anatomy in the paraaortic region is illustrated in Figure 1. Classical anatomy clarifies PALNs as lumbar lymph nodes, lumbo–aortal lymph nodes or lumbo–abdominal lymph nodes [6]. PALNs are defined based on their relationship to the abdominal aorta (AA) and the inferior vena cava (IVC). They are separated as left lumbar lymph nodes (*lumbales*
*sinistri*), aortocaval lymph nodes (*lumbales intermedii*) and right lumbar lymph nodes (*lumbales*
*dextri*). Additionally, PALNs are divided into the following three groups: left lumbar lymph nodes—lateroaortic (*aortici laterals*), preaortic (*preaortici*), retroaortic (*postaortici*); right lumbar lymph nodes—laterocaval (*cavales*
*laterals*), precaval (*precavales*), retrocaval *(postcavales*), and aortocaval, also known as interaortocaval (*lumbales intermedii*) (Figure 2, Figure 3 and Figure 4) [6,7,8].

## 3. Boundaries of PALND in Gynecological Malignancies

PALND could be separated into sentinel lymph node biopsy, excision of only enlarged nodes or systematic [9]. PALND can be performed by laparotomy, laparoscopy or robotic surgery and through either a transperitoneal or retroperitoneal approach [10]. In the present article, open systematic paraaortic lymphadenectomy by a transperitoneal approach will be discussed, as all the lymph nodes in the paraaortic field are removed. Therefore, boundaries and regions will be well demonstrated.

Boundaries of PALND are [11,12,13] (Figure 5):-Right—ureter, Gerota fascia, psoas major muscle, ascending colon;-Left—ureter, Gerota fascia, psoas major muscle, descending colon;-Ventrally—left renal vein;-Dorsally—midpoint of common iliac vessels;-Caudally—anterior longitudinal ligament.

There is no consensus about the ventral boundary of PALND in patients with cervical cancer. Some oncogynecologists stated that the ventral limit should be the inferior mesenteric artery (IMA), as PALNs metastases superior to the IMA are infrequent in cases of absence of node metastasis inferior to the artery [14,15]. Ouldamer et al. performed a systematic search in order to estimate the level (IMA or renal veins) of PALND in cervical cancer patients. Authors observed eight patients (1.09%) with isolated para-aortic node metastases. Two of them had lymph node metastases superior to the IMA [15]. However, in one of the biggest studies, 616 women with locally advanced cervical cancer underwent PALND up to the left renal vein (LRV). Results showed that 114 patients (18.5%) had PALNs metastases, of which 73 (64%) had supramesenteric metastases. Moreover, 11 patients (9.6%) had metastases only in the supramesenteric region [16]. Additionally, the presence of skip metastases should also be considered. [17]. Llueca et al. observed 394 patients with locally advanced cancer of the uterine cervix. Authors found positive PALNs in 119 women. Skip metastasis in the supramesenteric region was found in 18% of lymph nodes retrieved, without the presence of a metastases inframesenteric region [17]. According to these studies, we concluded that the LRV should be considered as the ventral limit of PALND for patients with cervical cancer and suspected PALNs’ metastases.

## 4. Regions and Their Boundaries during PALND

During PALND, the following regions are dissected [11,12] (Figure 6):

The paraaortic region (includes lateroaortic and retroaortic PALNs) is divided by the IMA into two regions.

(A)The high paraaortic (supramesenteric) region is limited: ventrally—LRV; medially—AA; laterally—ureter and Gerota fascia; dorsally—IMA; caudally—psoas major muscle.(B)The low paraaortic (inframesenteric) region is limited: ventrally by the IMA; medially—by the AA, dorsally—the left common iliac artery (CIA); laterally—the ureter and the Gerota fascia, caudally—the psoas major muscle.(C)The aortocaval or interaortocaval region (includes preaortic and precaval PALNs) is limited: ventrally—LRV, laterally—left—lateral aspect of the AA, right—lateral aspect of inferior vena cave, dorsally—AA bifurcation, caudally—prevertebral fascia, anterior longitudinal ligament and psoas major muscle.(D)The paracaval region (includes laterocaval and retrocaval PALNs) is limited: ventrally—right renal vein (RRV); dorsally—midpoint of the lateral aspect of right CIA, laterally—right ureter and right psoas major muscle, caudally—the psoas major muscle.

Usually, retrocaval and retroaortic PALNs are removed during PALND in the paraaortic and paracaval regions, so a particular region in that zone is not defined. The supramesenteric region includes infrarenal regions, as the limit of dissection in the supramesenteric regions is the LRV.

## 5. Pathways of Lymphatic Spread to the Paraaortic Region in Gynecological Pelvic Cancers

Pathways of lymphatic spread in cancer of the uterine cervix are still debatable. Ercoli et al. performed a macroscopic morphological study and distinguished three different lymphatic pathways—supraureteral, infraureteral and the neural paracervical pathways [18]. Kraima et al. reported two main pathways, which drained the uterine cervix—the first pathway passes through the cardinal ligament above the ureter, and the second continues in the sacrouterine ligament to the rectal pillars [19]. Cibula and Rustum described two major lymphatic trunks—superficial and deep stems. The superficial passes through the ventral wall of the external iliac artery/vein and CIA. It then continues cranially to the precaval and interoaortocaval region.

The deep trunk passes from the medial to the external iliac artery/vein and around the obturator nerve. At the level of the superior gluteal artery/vein and obturator internus muscle, it separates into two pathways, which drain in the laterocaval and interaortocaval, preaortic regions, respectively. Both trunks have numerous channels with the parametrium [9].

Pathways of lymphatic spread in endometrial cancer depend on the primary tumor locations. Usually uterine cancers, which is located at the lower and upper aspect of the uterus affects obturator lymph nodes. Metastases to PALNs are more likely to occur in uterine cancers located to the upper uterine corpus or uterine fundus. In such cases, there are two possible pathways. The first includes spread to the lymph nodes located near the internal and common iliac vessels and then to the PALNs. The second lymphatic pathway follows the gonadal vessels and directly continues the PALNs [7]. Therefore, PALNs are considered as regional and isolated PALNs metastases are a possible scenario in endometrial cancer.

Lymphatic ovarian vessels merge upon the ovarian hilus, where a sub-ovarian lymphatic plexus is formed. There are three pathways of lymphatic spread through the plexus. The main lymphatic truncus follows the ovarian veins to the IVC on the right side and the LRV on the left, respectively. The sentinel lymph nodes for the right ovarii are aortocaval and laterocaval chains (located at L1-L2), whereas for the left ovarii they are lateroaortic and preaortic lymph nodes (located slightly inferior to the LRV). Therefore, systematic PALND for ovarian cancer includes excision of the ovarian vessels at their origin. Although on the right side the ovarian vein drains in the IVC, the PALND proceeds to the level of the RRV [7,8,20]. However, there are also controversies in lymphatic pathways in ovarian cancer. Kleppe et al. examined tissue samples from three female fetuses and one adult cadaver and described three ovarian lymphatic drainage pathways—two major and one minor. The first major pathway passed through the ovarian ligament to the obturator and internal iliac lymph nodes, whereas the second passed through the infundibulopelvic ligament to the supra/infra mesenteric and paracaval regions. The third drained the gonads through the round ligament to the inguinal nodes [21].

Additionally, it should be noted that metastatic lymphatic spread also depends upon the histology of the main cancer. For instance, Takeshima et al. investigated 208 women with ovarian cancer who underwent complete pelvic and PALND. Authors concluded that, in case of a serous histology, the supramesenteric area was the main site for the earliest lymph node metastases [22]. Moreover, type 2 endometrial cancers (serous; clear cell) and cervical adenocarcinoma are associated with higher incidence of pelvic and PALNs metastases compared to type 1 cancers [2,3,4,5,23].

## 6. Systematic Paraaortic Lymphadenectomy Technique 

A xypho-pubic skin incision is performed for better exposure of the AA and IVC from their bifurcation up to the renal veins.Small bowels, omentum, colon transversum are exteriorized cranio-laterally and the sigmoid colon is retracted caudo-laterally.Two separate incisions of the posterior parietal peritoneum are performed: the first along the right paracolic gutter to the level of the hepatocolic ligament and the second incision along the ileocolic junction to the level of the ligament of Treitz.Entering the avascular plane between Gerota’s and Toldt’s fascia.Identification of the right ovarian pedicle and separation from the right ureter.The transversal part of duodenum is dissected superiorly.The areolar tissue is dissected between the left CIA and sigmoid mesentery. Identification of the IMA, left ovarian pedicle and left ureter is performed.Mobilization and lateralization of the ureters is performed. Ureteric vessels should be preserved in order to prevent fistulas.Ligation of ovarian vessels at their insertion.The dissection of PALNs is performed in the caudal to the cranial direction.Laterocaval, retrocaval and precaval lymph nodes are dissected. The presence of the lympho-vascular anastomosis draining into the IVC anteriorly should be considered.Preaortic lymph node dissection. Attention during dissection for the lumbar vessels. Ligation of the aortocaval lymphatic is performed.Lateroaortic and retroaortic lymph node dissection in the supra/inframesenteric region performed. It is recommended to avoid ligation or injury of the IMA, although its resection and ligation rarely causes ischemia. Ligation of suprarenal lymphatic vessels [8,11].

## 7. Complications of PALND

Intraoperative: hemorrhage, injury of lymphatic vessels, nerves and organs (bowels, kidneys and ureters) [13,24].

Postoperative: infections, sepsis, hemorrhage, lymphatic complications (lymphocysts, lymphorrhea), low extremity lymphedema, thrombosis, pulmonary embolism, prolonged ileus (commonly obstructive), fistulas, chylous ascites, and autonomic nervous system injuries [13,24].

Intraoperative bleeding is the most common complication during PALND due to iatrogenic vessel injury, poor tissue handling, exuberant retraction and possible anatomical variations of the vessels in the paraaortic region. Approximately, 30% of women will have at least one vessel variation in the paraaortic region [13]. Therefore, knowledge of anatomical vessels variations is mandatory in order to decrease morbidity and mortality.

## 8. Risks and Benefits of Systematic PALND

It should be stressed that the extent of the PALND required for an individual case should be determined by an oncogynecologists who is totally experienced and familiar with the disease (individual peri-operative management of patients and histological type, pathologic grade, molecular characteristics of the tumors) and the effectiveness of all available non-surgical therapies. Therefore, of utmost importance is whether the risks of a systematic PALND outweighs the benefits. Extent radicality is not always the key to better survival and may increase mortality. In order to decrease morbidity, and to improve oncologic care in women with ovarian cancer, guidelines for peri-operative management have recently been introduced [25]. Currently, studies which showed that less radicality is associated with similar outcomes compared to radical surgery have been widely accepted [24,26]. The lymphadenectomy in ovarian neoplasms (LION) trial proved that patients with normal lymph nodes, who underwent PALND, were not associated with increased overall or progression-free survival than patients in the no lymphadenectomy group. Moreover, more postoperative complications were observed in patients with the lymphadenectomy group [24]. Additionally, another study reported that completion of radical hysterectomy is not associated with increased survival in women with intraoperatively detected lymph node involvement, regardless of the tumor size or histology [26].

In conclusion, multidisciplinary team management and appropriate surgical selection criteria are needed in order to improve patient outcome.

## 9. Vessels Variations in the Paraaortic Region

### 9.1. Renal Arteries Anatomy

Left and right renal arteries (RAs) typically arise from the lateral aspect of the AA slightly inferior to the origin of the superior mesenteric artery and at the level of the intervertebral disk, located between L1 and L2 [27,28,29]. RAs supply approximately 25% of cardiac output to the kidneys [29]. The RAs are situated superior to the renal veins. The right renal artery (RRA) is longer than its left counterpart due to the AA topographic localization more to the left of the vertebral column [27,29]. The origin of the RRA is slightly higher than the left one [29]. The RRA typically passes behind the IVC, RRV and head of the pancreas. The left renal artery (LRA) passes behind the LRV, pancreas body and splenic vein [27]. The RAs divide just prior to piercing the renal hilum into anterior and posterior branches. The anterior branch additionally separates into the upper, middle, lower, and apical segmental arteries, whereas the posterior branch does not give any branches and forms the posterior segmental artery [30]. The paired inferior suprarenal arteries originate from the RAs [29]. Segmental arteries or any renal vessel variation located close to the kidney hilum will not be the topic of the present article, as they are not at risk of iatrogenic damage during PALND.

#### 9.1.1. Renal Arteries Anatomical Variations

Variations of the RAs are more frequent in the paraaortic region, and the incidence of variations shows gender and racial differences. They are more common in males and among African and Caucasian populations [31,32].

##### Type IA—Anatomical Variations of the Origin of Renal Arteries, Which Arise from the Abdominal Aorta

Özkan et al. performed angiographic research of nearly 900 patients and reported that the origin of the main RAs of the AA was between the upper margin of L1 and the lower margin of L2 vertebra in 98% of the cases. They also observed that 23% of right and 22% of the left RAs arose at the intervertebral disk between L1 and L2. Other similar studies supported these findings [31,32]. Most studies concluded that the majority of RAs arose between the L1, L2 vertebrae or at the level of the disk between them [28]. However, there are cases of RAs arising from the upper or lower part of the AA. Baalen and Bockel observed a case of a women with an RRA, which arose at the level of the eleventh thoracic vertebra superior to the celiac trunk [33]. Other case reports showed the same origin variations [34,35]. The incidence of the renal artery (RA) originating at the level of the celiac trunk is approximately 3% [33,36].

##### Type IB—Anatomical Variations of the Origin of Renal Arteries, Which Arise from the Abdominal Aorta Branches

In these particular cases, RAs, which arise from the AA branches, will be discussed, as they have a different course in the paraaortic region and their injury during PALND is more likely to occur. Halloul et al. reported a case of RRA originating from the left CIA [37]. Kryut reported a patient with a sole LRA originating from the splenic artery. The diagnosis of the anatomical variation was achieved by angiography and confirmed by surgery [38]. Cases of common origin of one of the RAs and IMA have also been described [39,40]. Bartoli et al. reported a case of a 38-year-old woman with a celiomesenteric and LRA originating from a common stem [41]. Incidence of Type IB RA variations are unknown, but it is stated that they are infrequent, as only a few cases have been reported [37,38,39,40,41].

##### Surgical Considerations during PALND

The surgeon should be aware of RA origin variations, which arise from the AA, in order to avoid the unexpected injury of these arteries during PALND. Regarding the RAs, in some cases they may be seen below the level of renal veins rather than their proper localization between the L1 and L2 vertebra, and the surgeon should dissect the PALNs located at the level of renal veins very carefully in order to prevent an injury of the RAs (Figure 7). RRA originating from the left CIA had important surgical implications, as injury during surgery may cause ischemic necrosis of the kidney. Moreover, ligation of IMA closed to the AA may lead to kidney necrosis in cases of common trunk with the LRA. Therefore, ligation of IMA during PALND should be avoided even in cases of bulky lymph nodes. Injury of Type IA and IB renal variations has not been described in gynecologic oncology surgery, but Mazzeo et al., reported iatrogenic injury of LRA during left laparoscopic hemicolectomy. Authors observed variant origin of the LRA—more inferiorly than usual compared to the normal population [42].

#### 9.1.2. Additional and Accessory Renal Arteries

Many terms have been used to define these RA variations, including ‘aberrant’, ‘abnormal’, ‘accessory’, ‘additional’, ‘multiple’, ‘supernumerary’ and ‘supplementary [43,44]. Consequently, the discrepancy in nomenclature confuses surgeons and anatomists.

Therefore, the widely adopted term “additional renal artery” proposed by Satyapal et al. will be used throughout the present article. An additional RA is another artery apart from the primary RA, which originates from the AA or RA and ends in the kidney. Double or triple RAs are defined as first and second additional RA [33]. The term “accessory renal artery” will be used for all renal RAs than the main RAs, which arise from the aortic branches (except RA) and terminates in the kidney [28,43]. Moreover, additional and accessory RAs could be divided into hilar or polar according to their entrance to the kidneys—into the hilus or directly into the capsule outside the hilus [28,43,44].

##### Additional Renal Arteries

The incidence of additional RAs varies from 10% to 50%, but most studies indicated that approximately 30% of the population would have at least one additional RA [29,43,44,45,46].

Satyapal et al., analyzed 440 kidneys during cadaver dissection, surgery and angiograms. Authors reported the incidence of additional RAs and first additional RAs to be 23.2% and 4.5%, respectively. Additional left RAs were more common. The incidence of bilateral additional RAs was 10.2%. An Additional RA is more common among the male population [43].

Saldarriaga et al. investigated 196 renal blocks. An additional RA was found in 22.3% of the blocks and the first additional ones were found in 2.6% of the same sample. The left additional RA was the most common variation [45].

Dogra et al. observed 100 patients through CT angiography on a 64-slice MDCT scanner. Additional RAs were observed in 36% of cases with unilateral anomaly in 30% of cases. Only 6% of the examined patients had bilateral additional RAs. Contrary to Satyapal and Saldarriaga findings, authors reported that the ride side was the dominant side of RA variations [46].

Tardo et al. analyzed the RAs of 594 kidneys from 300 subjects (28 cadavers, 272 CT). Additional RAs were observed in 22% of subjects and 12.12% of kidneys. There were no differences between genders. First and second additional RAs were discovered in 5.6% and 1.4% of subjects with RAs variations [44].

The incidence of second additional RAs is approximately 2% [28].

As mentioned above, additional RAs could be separated into polar and hilar. Additionally, polar arteries could be divided into aortic upper polar, aortic lower polar, renal upper polar and renal lower polar, as they could arise from the AA or the RA. Ravery et al. reported that lower polar RA occurred in 9% of the population [47]. In a cadaveric study, Kommuru et al. observed 184 kidneys. The presence of aortic upper polar RA was found in 25% cases and aortic lower polar RA in 31.5% of cases [48]. Generally, the incidence of aortic lower polar RA is approximately twice that of aortic upper polar RA [31]. Additional hilum RAs are more common than polar ones [44,45,46]. First and second additional RAs could have a polar and hilar entrance into the kidney [28,44,45,46,48]. An additional RRA mostly arises from the anterolateral surface of the AA or sometimes from the right CIA. In cases of additional right RAs, vessels often pass anterior to the IVC (Figure 8 and Figure 9) [28,44,45,46,49]. Jeffery reports a case of a 65-year-old woman with additional LRA arising from the LRA and entering the lower pole of the right kidney. It had a preaortic and precaval course [50]. Moreover, our experience and the observation of the majority of authors demonstrated that the additional LRA typically arises from the left lateral aspect of the AA (Figure 10 and Figure 11) [28,44,45,46,49].

##### Surgical Considerations

An additional LRA could be injured during dissection in the infra/supramesenteric regions, whereas iatrogenic damage to the additional RRA is more likely to occur in the aortocaval and paracaval regions. Moreover, during PALND in the infra/supramesenteric region, the left ureter should be dissected under the mesentery of the sigmoid colon after detaching the peritoneum of the mesentery from the anterolateral aspect of the AA (Figure 12). Additional aortic lower polar RAs seem to have greater clinical significance than additional aortic upper polar and hilar arteries. Additional aortic lower polar RA gives a branch to the superior part of the ureter. Iatrogenic injury to such an artery will cause ureteric necrosis, fistulas and urinary leak. Moreover, such an artery may cause a ureteric obstruction and potential hydronephrosis. During surgery, oncogynecologists should be aware of this vessel’s variation, if they observe hydrouereter without obvious cause for its occurrence [44]. Furthermore, ligation of additional aortic polar RAs is associated with segmental ischemia and failure of the kidney, as these arteries are end arteries and cause an infarction of approximately 30% of the renal parenchyma [46,47,48,49,50,51]. Eitan et al. reported an additional aortic lower polar RRA injury during PALND for endometrial cancer. Its damage caused an infarction of the lower pole of the right kidney. The patient had a normal postoperative course [51]. Benedetti-Panici et al. reported one ligation of an additional aortic lower polar RRA among 309 operations for gynecological malignancies. The woman had no postoperative complications [52]. Although oncogynecologists often described additional RAs during PALND, the real incidence of injury is unknown [53,54]. However, all authors concluded that surgeons should be familiar with these potential anatomical variations [28,29,30,31,32,33,34,35,36,37,38,39,40,41,42,43,44,45,46,47,48]. Classification of additional RAs is shown in Figure 13.

##### Accessory Renal Arteries

Accessory RAs could originate from all of the AA branches [44,46,55,56]. Accessory RAs could also arise from the common iliac arteries [46]. Gulsun et al. observed a female patient with three accessory RAs. The three arteries entered into the right kidney, which was located in the pelvis—two from the left and right common iliac arteries and one from the ipsilateral internal iliac artery. Authors established these vessel variations by angiography [57]. As additional RAs, the accessory RAs could be hilar, polar, first and second [28].

##### Surgical Considerations

Surgeons should be aware of accessory RAs, as they have different courses through the paraaortic region. The case of Gulsun et al. is not associated with vessel variations in the paraaortic region, but it is mentioned as ectopic kidney is often associated with the presence of accessory RAs [57,58]. Moreover, these variations are located in the dorsal limit of PALND.

#### 9.1.3. Precaval Right Renal Artery

Generally, the RRA passes behind the IVC, but it can also have a precaval course. The incidence of precaval RRA varies between 0.8% and 5%. Srivastava et al. made a retrospective study, which included a 73-contrast enhanced multidetector CT angiography. Authors reported prevalence of precaval RRA of 5.48%. They concluded that the incidence of precaval RRA appeared to be more frequent than previously thought [49]. Yeh et al. came to the same conclusion as they reported incidence of 5% of the artery. Authors stated that rotation of the lower pole of the right kidney in the anterior direction is associated with higher rate of precaval RRA [59].

##### Surgical Considerations

Iatrogenic damage to the precaval RRA is possible during PALND in the aortocaval and paracaval regions. Incidence of injury of precaval RRA is unknown. It is possible for such a variation to confuse surgeons during dissection. Kose reported injury of LRV in a woman with precaval RRA during PALND [60].

### 9.2. Renal Veins Anatomy

Renal veins (RVs) are situated anterior to the RAs and enter the IVC at the level of the L2 vertebra. The LRV is three times longer than the right in length (7.5 and 2.5, respectively). The RRV is located behind the second part of the duodenum. There are no veins which drain in the RRV. The LRV has a preaortic course and enters in lateral aspect of IVC. Generally, the left suprarenal vein and left ovarian vein drain into the LRV [23,49]. The superior mesenteric artery is located above the LRV [23].

#### 9.2.1. Renal Veins Variations

RV variations are classified into: additional RVs, circumaortic RV, retroaortic LRV and retropelvic tributary of the RV [61,62,63,64]. Variations such as late venous confluence and plexiform LRV are not included, as these variations are close to the renal hilum, and injury is less likely to occur during PALND [62,64]. Anatomical variations of the LRV have great clinical significance during PALND.

#### 9.2.2. Variations of the Draining Pattern of the Left Renal Vein

The LRV could have different draining patterns in the IVC. Such a draining pattern variation has a great oncological significance; as mentioned above, the LRV is the ventral boundary of PALND. Kose et al. demonstrated incidence of 3.9 of LRV entering in the IVC distally, at the origin of the IMA [60].

##### Surgical Considerations

In cases of distal LRV entering the IVC, the ventral border of the PALND should be the right ovarian vein, where it terminates into the IVC [60].

#### 9.2.3. Additional Renal Veins

Additional RV is defined as any other vein than the main vein, which arises from the hilum of the kidney and enters the IVC [64]. The most frequent variation of the RRV is an additional vein, which is approximately 20–23% [64]. The frequency of additional RVs is much higher on the right than on the left side [62]. In a meta-analysis, Hostiuc et al. reported an incidence of additional RRV and LRV of 14.2–19.1% and 1.3–3.2%, respectively [62]. Satyapal et al. also reported prevalence of additional RVs on the right sight. Authors reported the incidence of additional RVs as 26% on the right side and 2.6% on left sight [43]. Additional LRV could a have preaortic course [44,63,64,65]. Additional RVs may arise from the hilum or polar of the kidney before draining into the IVC [63]. First, additional RVs are not common findings on the right side (5%) [43].

##### Surgical Considerations

Iatrogenic damage of additional RVs is possible during dissection of all the paraaortic regions. The presence of additional LRV raises a question—which of them is the ventral boundary of dissection, especially if the additional LRV is located superior to the main LRV? We concluded that the main LRV is the boundary of PALND and further dissection is not necessary.

#### 9.2.4. Circumaortic Left Renal Vein

The circumaortic left renal vein (CLRV), also known as ‘‘circumaortic venous ring’’, or ‘‘renal collar’ involves vessels variations, where an additional RV, apart from the main LRV, has a retroaortic course and drains in the IVC or the left common iliac vein (CIV) [6] (Figure 14). Some authors define different types of CLRV according to the draining pattern of the main LRV and CLRV. We believe that these types are not relevant during PALND [62,65]. The clinical significance of CLRV during PALND depends on the additional vein, which joins either the IVC or the left CIV, as these vessels have different courses through the paraaortic region. Hostiuc et al. estimated an incidence of CLRV of 3.5% [62,65]. Yi et al. analyzed that the median rate of CLRV was 7.0% in cadavers and 1.8% in clinical subjects, respectively [63]. Kuzan et al. observed left RV variations among 12,341 patients examined by CT. The frequency of CLRV was 0.70%. Authors concluded that there was no difference in incidence of CLRV between women and men [66].

##### Surgical Considerations

Surgeons should be careful during dissection in the aortocaval region in cases of CLRV draining in the IVC and in the supra/inframesenteric regions in cases of CLRV, which drains into the left CIV.

#### 9.2.5. Retroaortic Left Renal Vein

LRV, which passes posterior to the AA, is called a “retroaortic left renal vein” (RLRV). RLRV may drain into the IVC or the left CIV [64]. RLRV may have an orthotopic position or drains into the IVC at the level of L4–L5 [67]. Hostiuc et al. reported an incidence of 3% for the RLRV [62]. Another study reported an incidence of 1.7% of RLRV in cadavers [63]. Other studies reported prevalence of 1.84% and 6.6% for RLRV [66,68]. Cases of double RLRV have also been described [69,70].

##### Surgical Considerations

Surgeons should be aware of RLRV during dissection in the supra/inframesenteric and aortocaval regions. Moreover, the presence of one RLRV does not exclude the existence of another. Additionally, injury to the superior mesenteric artery is possible in cases of RLRV, as the surgeon may extend the cranial limits of dissection in order to try to find the absence preaortic LRV (Figure 15).

#### 9.2.6. Retropelvic Tributary of the Left Renal Vein

A retropelvic tributary (RPT) of the LRV, also termed the posterior renal vein or the supernumerary renal vein, is a frequent vessel variation [63,71]. Satyapal et al. observed a rate of 23.2% in a cadaveric series [72]. Yi et al. found that a rate of RPT of the renal vein being 30.0 to 46.4% [63]. Additionally, authors observed high frequency of anastomotic veins between the LRV and the veins of the retroperitoneum (hemiazygos, lumbar etc.), which ranged from 30.0 to 84.2% in cadavers. Authors proposed that RPT of the LRV should not be considered as a variation but a normal feature due to its high incidence [63].

##### Surgical Considerations

RPT of the LRV could be injured during infra/supramesenteric lymph node dissection.

##### Incidence of Iatrogenic Injury of RVs Variations

Panici et al. described variation in the retroperitoneum and associated surgical complications that were observed during lymphadenectomy in 309 patients with gynecological cancers. Authors found additional polar RV and CLRV in 0.6% and 1% of cases, respectively. No injury of these anomalous vessels was reported [52]. Kose et al. analyzed retroperitoneal vascular variations during oncogynecological retroperitoneal surgery among 229 women [60] Authors observed additional RVs and RLRV in 0.9% and 3.1% of cases. They reported for injury of LRV in a patient with RLRV and another injury of LRV, which drained distally to the IVC. Moreover, authors caused iatrogenic injury of right CIV in a patient with additional RRV [60]. Gyimadu et al. investigated the frequency of retroperitoneal large vessels variations in patients with gynecologic malignancies, who underwent pelvic and PALND. Authors observed vessels variations in 24.3% of patients by multi-detector computed tomography. CLRV and RLRV were observed in 8.1% and 2.7% of patients. Six patients (16.2%) had vascular intraoperative injuries during lymphadenectomy. Although injury of RVs variations did not occur, authors concluded that the rate of intraoperative vascular injuries was more common among patients who had vessels variations [73]. Truty et al. observed vessels variations among 2400 patients, who underwent abdominal and thoracic aortic aneurysm reconstruction [74]. The incidence of CLRV and RLRV was 0.16% and 1.08%, respectively [74]. Iatrogenic injury of CLRV and RLRV occurred in 50% and 19% of patients with these vessel variations [74]. Seyfettinoglu et al. reported a ligation of RLRV during PALND in a patient with ovarian carcinoma. Authors manage to save the left kidney by a gonadal vein graft to maintain renal circulation [75]. Although the number of cases with injury of RVs is limited, it is obvious that the presence of vessel variations is related with higher incidence of iatrogenic vessel injury.

### 9.3. Abdominal Aorta Variations

AA variations are mostly associated with their branches, which will be described separately. Although major variations of AA are rare, cases of double AA have been described [76,77]. However, iatrogenic injury to double AA is less likely to occur during PALND. The presence of recently described the anterior retroperitoneal branches of the AA should also be stressed. Turyna et al. examined 25 AA samples with the surrounding tissue from the retroperitoneum. Samples, which included both sexes, were dissected by using magnifying binocular glass and graphic reconstruction. Authors found minute arteries arising from all levels of the AA. Authors separated the AA into three levels according to the origin of RAs and inferior mesenteric artery. These arteries were termed “anterior rami of the abdominal aorta”. According to Turyna et al., these arteries, though thin, could be confused with ovarian arteries or accessory RAs. Therefore, surgeons should be familiar with their presence during PALND [78].

### 9.4. Inferior Vena Cava Anatomy

The IVC carries blood to the right atrium from the majority of anatomical structures, which are located below to the diaphragm. It arises from the junction of the left and right common iliac veins at the level of fifth lumbar vertebra. IVC has four segments: hepatic, suprarenal, renal and infrarenal. The AA is located lateral to the left side of the IVC. It is covered by the parietal peritoneum and the third part of the duodenum. IVC has multiple tributaries: lumbar veins, right ovarian and suprarenal, inferior phrenic, and veins of the liver [27,79].

#### 9.4.1. Left Sided Inferior Vena Cava

The left sided inferior vena cava (LIVC) is the second most frequent variation of the IVC [80]. The LIVC is a result of abnormal regression of the right supracardinal vein with persistence of the left supracardinal vein. The LIVC is a mirror image of normal anatomy—the right ovarian and adrenal veins enter the RRV, whereas the left counterparts terminate in the IVC [81]. There are two different types of LIVC. In Type I, the IVC passes behind the level of the LRV to form a normal right-sided IVC. In type II, the IVC continues its passage upward, without crossing the AA [80]. The LIVC occurs in 0.2–0.5% of the population (Figure 16 and Figure 17) [81,82].

##### Surgical Considerations

The most important significance of LIVC during PALND is the possibility of confusing these variations with lymphadenopathy, tumor or a dilated ovarian vein [80,81,82]. Therefore, dissection of the supra/inframesenteric regions should be meticulous.

#### 9.4.2. Duplication of the Inferior Vena Cava

The duplication of inferior vena cava (DIVC) was first described by Lucas in 1916 [79,83,84]. Since then, numerous case studies and proposal for classification of DIVC have been described [81,82,84,85,86]. We propose a classification according to the DIVC clinical significance during PALND in gynecologic oncology. DIVC could be separated into DIVC with regressed right IVC, right-sided DIVC and complete bilateral DIVC (BDIVC) [85,87,88,89]. Homma et al. reported a case of DIVC—left inferior vena cava (LIVC) with regressed right IVC. The regressed IVC received tributaries from the right third lumbar vein and posterior renal veins, and from a vein, located in the inferior mesenteric plexus [87]. Up to now, no more than 10 cases of LIVC with regressed right IVC have been reported [28,87]. However, in our practice we also observed an LIVC with a regressed right IVC and multiple presacral venous communications between both caval systems (Figure 18).

The right-sided DIVC is defined as a presence of two infrarenal right IVCs, which are located at the right side of the AA. Tagliafico et al. described a case of a right-sided DIVC with an anomalous venous ring. [88]. Nagashima et al. described five cases of right-sided DIVC encountered in CT imaging findings. Authors concluded that the ventral vein had a larger diameter than the dorsal vein and the right gonadal vein usually drains into the ventral vein. Right-sided DIVC is a rare variation, as approximately 10 cases have been described so far [89].

BDIVC is the most common variations among the IVC variations [85]. The incidence of BDIVC ranges between 0.2% and 3% with intraoperative findings between 0.2% and 0.6% [79]. BDIVC is caused by a failure of regression of the left supracardinal veins during embryogenesis [79]. The LIVC typically has a preaortic course and drains into the LRV, which has a preaortic course [13]. The LIVC could also drain in the right IVC and the LRV drains in the LIVC [90]. Cases of LIVC with retroaortic course, which drains into the right IVC, have been described [81]. Many classifications have been introduced for better understanding of BDIVC [81]. Morita et al. proposed a classification of BDIVC based on the pattern of interiliac communication [86,91]. According to Natsis, these variations are classified according to size of the caliber trunks of both IVCs and the preaortic trunk [85]. Our classification is based on draining patterns of DIVC and large venous anastomoses between both IVCs. We believe that these variations have clinical significance during PALND. Therefore, hemiazygos continuation of the IVC is not mentioned, as it is not in the operation field during lymphadenectomy. Classification of the DIVC is shown in Figure 19. Cases of BDIVC are shown in Figure 20 and Figure 21 [83,84,85,86,87,88,89].

##### Surgical Considerations

DIVC could be misdiagnosed with lymphadenopathy in the paraaortic region, accular aortic aneurysms, dilations of the ureter, loops of small bowel or cysts in the retroperitoneum [92,93,94]. Therefore, surgeons should be aware of the possible presence of DIVC in such cases. BDIVC with interiliac anastomoses could be injured during PALND in all paraaortic regions. During lumboaortic lymphadenectomy for patients with ovarian cancer, Gomes et al. observed BDIVC with communications between both common iliac veins. Authors concluded that these variations might cause serious hemorrhage, especially from the interiliac anastomosis [91]. Kose et al. reported a rate of 17% of vessels variations among 229 women. Variations were encountered during lymphadenectomy in gynecologic oncology. Injury of the BDICV occurred in the only patient with that variation [60]. Gyimadu et al. observed 37 patients with vessel variation by preoperative multi-detector computed tomography (MDCT) followed by retroperitoneal surgery for gynecological malignancies. Authors found one case of BDIVC. The variant vessel was not injured due to the preoperative diagnosis by MDCT [73]. Ureyen et al. reported three cases of BDIVC found during PALND for gynecological cancers. In one of the cases authors injured the BDIVC, which was primary repaired and provided vascular continuation [90]. The actual incidence of DIVC iatrogenic damage in gynecologic oncology is unknown. However, injury to BDIVC was also reported in other specialties. Mao et al., reported iatrogenic injury of BDIVC during radical nephrectomy and cystectomy. Authors misdiagnosed the LIVC with the gonadal vein [95].

#### 9.4.3. Marsupial Inferior Vena Cava

Marsupial inferior vena cava (MIVC), also known as the preaortic iliac confluence, is a congenital variation, where the IVC or the left CIV are located before the bifurcation of the AA or the right CIA. Filho et al. reported MIVC mimicking lymph node enlargement on CT [96]. The variation is caused by ventral portion persistence of the embryologic circumaortic venous ring and dorsal portion regression of the ring [97]. It is labeled “marsupial”, as such an anterior position of the IVC is common among marsupials. The incidence of MCIS is unknown. Rocha et al. stated that 17 cases of MIVC have been found in the medical literature and the majority of them were observed during surgeries or autopsies. Therefore, approximately 20 cases have been described to date [96,97,98]. However, Rocha et al. concluded that such a variation could be not as uncommon as previously though, as the four cases which authors described were observed by the same radiologists [98].

##### Surgical Considerations

Surgeons and radiologists should be aware of MIVC, as it could mimic other conditions in the retroperitoneum. Iatrogenic injury to MIVC may cause life-threatening hemorrhage.

### 9.5. Ovarian Vessels—Ovarian Arteries Anatomy

Ovarian arteries (OAs) are branches of the AA, which generally originate below the RAs. They travel behind the parietal peritoneum, and at the brim of the pelvis cross the external iliac artery and vein to enter the pelvis. OAs give branches to the uterine tube, round ligament and anastomoses with the uterine artery [27,28].

#### 9.5.1. Ovarian Artery Variations

##### Variations in the Level of Aortic Origin and Position

There are three types of OA origins [28,99].

Type I—OAs arise from the AA inferior to the RVs and descend behind the peritoneum. The right ovarian artery (ROA) has a precaval course.

Type II—the ROA originates from the AA above the RRV and passes behind the IVC and in front of the RRV. The left ovarian artery (LOA) arises from the AA above the LRV and passes in front of the LRV.

Type III—The ROA arises from the AA, below the LRV, by arching over the RRV. The LOA arises from the AA, below the LRV, continues below and arches over the LRV.

##### Origin Variations of the OAs

OAs could either arise from the renal, iliac, lumbar, iliolumbar or from the additional RAs, either apart or from a common stem [28,100,101,102,103].

### 9.6. Ovarian Veins Anatomy

The ovarian veins (OVs) emerge as a plexus in the mesovarium and follow the course of the OAs. The left ovarian vein (LOV) enters the LRV, whereas the right ovarian vein (ROV) enters the IVC [27,28].

#### 9.6.1. Ovarian Veins Variations

OVs variations are rare compared to the variations of the OAs [29]. The most common variation among OVs is ROV draining into the RRV. The estimated incidence is 8.8% [28,104]. Ghosh and Chaudhury analyzed OVs variations in 94 sides of 47 embalmed cadavers. They reported of two cases of ROV variations—in one case, the ROV joined the RRV and, in the second case, the ROV was duplicated and drained into the IVC at two different locations. They reported a single case with a duplicated LOV. The three variations of the OVs were unilateral [105]. Benedetti-Panici et al. reported incidence of 2.2% for the LOV entering the lumbar vein. The frequencies of ROV entering in the RRV and ROV draining into the IVC above the RRV were 1.3% and 0.3%, respectively [52]. Koc et al. examined the ROV drainage variants of 324 women by abdominal multidetector-row computed tomography scans. Authors reported prevalence of 9.9% of ROV draining into the RRV [106].

#### 9.6.2. Surgical Considerations

Surgeons should be familiar with the draining pattern variations of the OVs, as during PALND the OVs should be ligated at their insertion into the LRV and the IVC.

### 9.7. Inferior Mesenteric Artery Anatomy

IMA originates from the anterior or left anterolateral aspect of the AA at the level of the third vertebra, or approximately 3–4 cm above the bifurcation of the AA [27,28]. Its branches are the left colic artery, sigmoidal arteries and the superior rectal artery [28].

#### 9.7.1. Inferior Mesenteric Artery Variations

Many classifications of IMA variations have been proposed, but most of them include branching pattern variations. These variations are not related to PALND, as their significance is greater in rectal surgery. Variations of IMA, which are related to PALND, include predominantly its origin variations. Balcerzak et al. analyzed IMA variations in 40 fixed cadavers. Authors found that IMA origin varies between the second and the fifth vertebra, but the most common origin was at the level of the third vertebra (35%) [107]. Zebrowski et al. reported approximately the same findings. In the majority of cases (67%), the IMA arose at the level of the third vertebra [108].

IMA could originate from the superior mesenteric artery [109]. Lippert and Pabst reported the frequency of such variations to be less than 0.1% [110]. Double inferior mesenteric arteries may also arise independently from the AA [111]. The IMA and additional RA can originate as a common stem from the AA [13,28]. Sher et al. described a case of IMA arising from the CIA [112]. Although extremely rare IMA could be absent [113].

#### 9.7.2. Surgical Considerations

It is advisable to preserve the IMA during PALND, as other vessels such as RAs could arise from the IMA. The absence of IMA does not exclude its presence, as it could arise from other vessels of the AA. However, if iatrogenic damage to the IMA occurs, it is not associated with any clinical consequences (if there are not vessels variations). The presence of two marginal arterial arches between the middle colic artery and the left colic artery prevent ischemic bowel complications after IMA ligation. The arches are known as the artery of Drummond and the Riolan’s arch [114,115].

The inferior mesenteric vein is not related to the present article, as it is not encountered during PALND.

### 9.8. Lumbar Arteries Anatomy

There are four pairs of lumbar arteries (LAs), which arise from the posterolateral aspect of the AA. The LAs pass posterolateral to the upper four lumbar vertebral bodies and continue into the abdominal wall muscles [27,116]. The right LAs have a retrocaval course [116]. The first pair of LAs arises between the celiac stem and the superior mesenteric artery. The next pair is located behind the RA. The third arises between the OA and the IMA, whereas the fourth originates inferior to the IMA, respectively [28].

#### 9.8.1. Lumbar Arteries Variations

The number of paired LAs is variable—between one and five [28,116]. Karunanayake and Pathmeswaran analyzed the variations of LAs in 109 cadavers. Authors observed that the number of left LAs varied from three to five, whereas the number right LAs varied from three to four. Authors also observed that, in 12% of cadavers, LAs arose from a common stem [116].

#### 9.8.2. Surgical Considerations

Iatrogenic injury of LAs is more likely to occur during PALND in the aortocaval regions. Surgeons should be aware that the number of LAs varies.

### 9.9. Lumbar Veins Anatomy

There also four pairs of lumbar veins (LVs), which drain into the IVC and collect the blood from the lumbar muscle, and skin. LVs anastomose with the azygos and hemiazygos vein. The left LVs are longer as they passed behind the AA. The first and second LVs may drain into the ascending lumbar vein and lumbar azygos vein. The first lumbar vein (LV) often joins the second LV [27,28].

#### 9.9.1. Lumbar Veins Variations

The most important significance of LVs variations is that they could drain into the LRV. Klemm et al. analyzed vessels variations in 86 women, who underwent PALND for gynecological malignancies. Authors found that venous variations occurred in 30.2% of patients. The most frequent venous variation was LVs, which drained into the LRV (17.4%) [117]. Baniel et al. investigated lumbar vessels in 102 patients, who underwent PALND. Authors found a high incidence of variations in the number and locations of the left LVs. The prevalence of LVs draining into the LRV was 43% [118].

#### 9.9.2. Surgical Considerations

As the majority of LVs variations are located at the left lateral aspect of the IVC, surgeons should be careful during PALND in the supra/inframesenteric and aortocaval regions.

## 10. Preventing Iatrogenic Injury of Variant Vessels

The majority of studies about vessels variations in the paraaortic region include cadaveric, intraoperative and imaging examinations. As vessels variations could be accessed by imaging modalities, careful preoperative evaluation of vessels variations is recommended. 3D CT angiography is the preferable imaging modality. Preoperative identification is important not only for PALND in gynecologic oncology but also for spinal surgery, general surgery and transplantation surgery [60]. Duan et al. reconstructed three-dimensional CT model of the pelvic vessels preoperatively in 30 women with cervical cancer scheduled for lymphadenectomy. Authors concluded that reconstruction of 3D models of the vessels in the pelvis could be very useful for identifying vessel variations [119]. A preoperative collaboration between surgeons and radiologist is necessary for identification of such variations.

It should be stressed to surgeons that there is possibility of false-negative results on imaging. False-negative results are less likely to be observed in large vessels such as the IVC and the AA. However, there are case studies reporting a preoperative imaging failure to recognized bilateral duplication of the IVC [91,120]. Moreover, as it was mentioned above, variant IVC could be misdiagnosed with different retroperitoneal pathologies [93,94,95]. In contrast, the false-negative results are higher for small vessels variations, especially if the vessels are occluded and nonfunctional. In our opinion, it is dangerous for the surgeon if renal vessels variations are not recognized preoperatively on CT or other imaging modality. Hassan et al. examined 56 patients for renal vessel variation by a multi-detector computed tomography angiography. The protocol of the study acquired CT data in all phases (arterial, venous and delayed). Authors observed 5.4% of false negative results in renal veins variations, whereas for arterial variations the result was 1.8%, respectively [121].

As the incidence of false-negative results is not low, intraoperative ultrasound (IOUS) may be of an additional value to find and better define vascular abnormalities, especially in the renal region. Currently, IOUS has been widely used in gynecologic oncology—assessment of myometrial invasion, avoiding unnecessary PALND in women with endometrial cancer, detection and localization of recurrent gynecological cancer, etc., [122,123,124]. Although there is no available study in gynecologic oncology describing IOUS and anatomical variations, in general surgery there are many examples [125,126]. Therefore, in our opinion, IOUS could be a useful tool for detecting suspected anatomical variations in gynecologic oncology.

Vessel variations in the paraaortic region and their clinical significance in Gynecologic Oncology are summarized in Table 1 [43,44,45,46,47,48,49,50,51,52,53,54,55,56,57,58,59,60,61,62,63,64,65,66,67,68,69,70,71,72,73,74,75,76,77,81,82,83,84,85,86,87,88,89,90,91,92,93,94,95,96,97,98,100,101,102,103,104,105,106,110].

## 11. Conclusions

The existence of vessel variations in the paraaortic regions is not uncommon as well as these variations significantly increase the risk of iatrogenic injury during PALND in gynecologic oncology. It must be considered that every patient who will undergo an operation may have vascular variations. Knowledge of renal vessels and IVC variation are of the upmost importance, as injury of these vessels is often associated with kidney infarction or life-threatening hemorrhage. Variation of some of the vessels may change the boundaries of PALND such as anomalies in the LRV. The awareness of these variations, an adequate exposure and a meticulous PALND dissection are vital to preventing injury. Preoperative and careful imaging examination by oncogynecologists and radiologists will help to decrease morbidity.

## Figures and Tables

**Figure 1 jcm-11-00953-f001:**
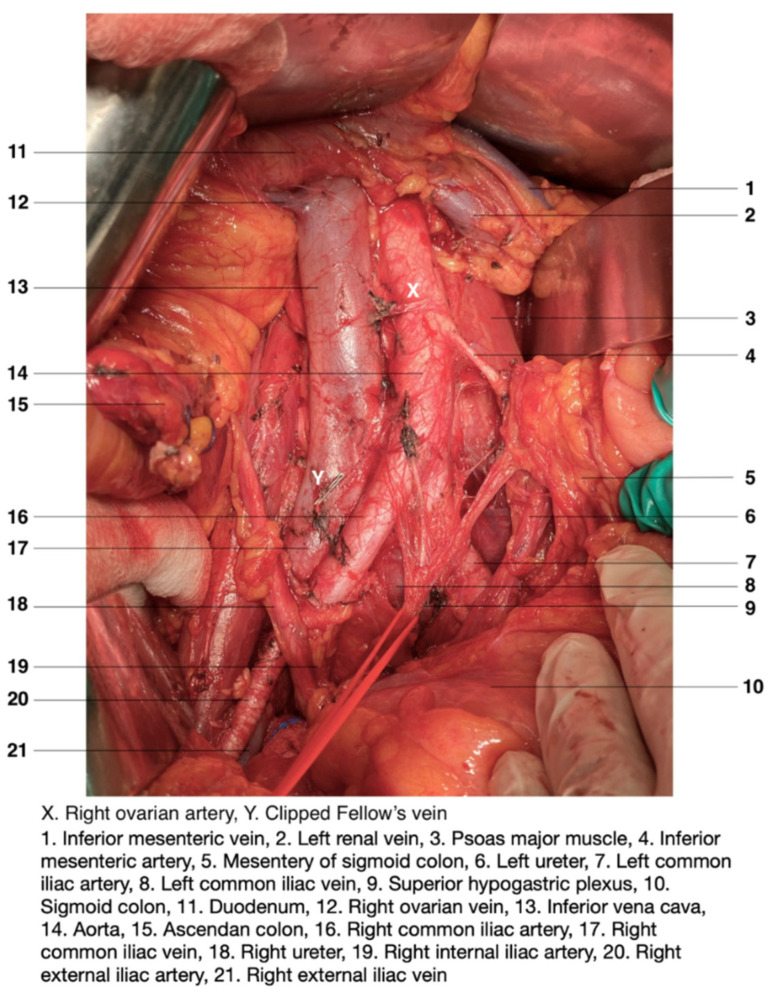
Topographical anatomy in the paraaortic region. An open surgery performed by doctor Ilker Selcuk.

**Figure 2 jcm-11-00953-f002:**
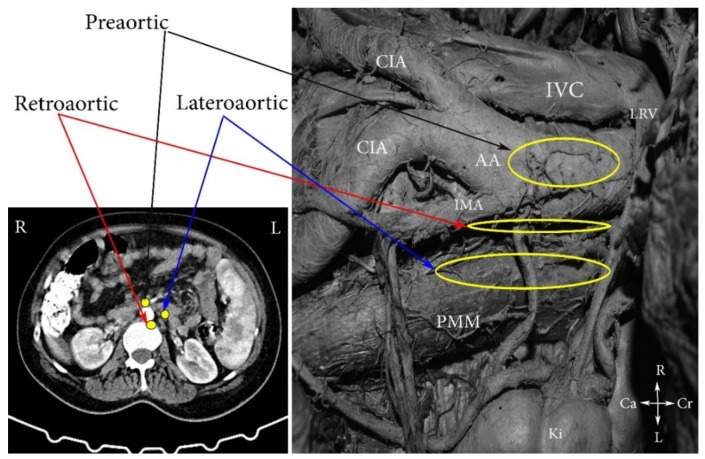
Left lumbar lymph nodes (embalmed cadaver). CIA—common iliac artery, IVC—inferior vena cava, AA—abdominal aorta, PMM—psoas major muscle, LRV—left renal vein; IMA—inferior mesenteric artery, Ki—kidney, Cr—cranial, Ca—caudal, L—left, R—right.

**Figure 3 jcm-11-00953-f003:**
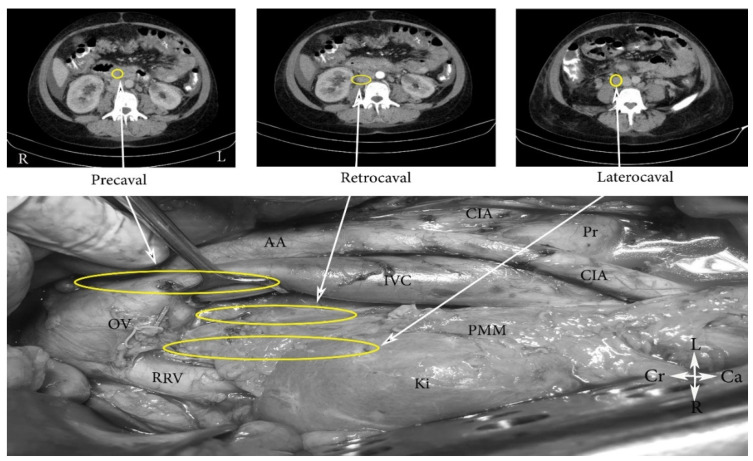
Right lumbar lymph nodes (open surgery). CIA—common iliac artery, IVC—inferior vena cava, AA—abdominal aorta, PMM—psoas major muscle, RRV—right renal vein; Pr—promontorium; OV—ligated right ovarian vein at its insertion to the IVC; Ki—kidney, Cr—cranial, Ca—caudal, L—left, R—right.

**Figure 4 jcm-11-00953-f004:**
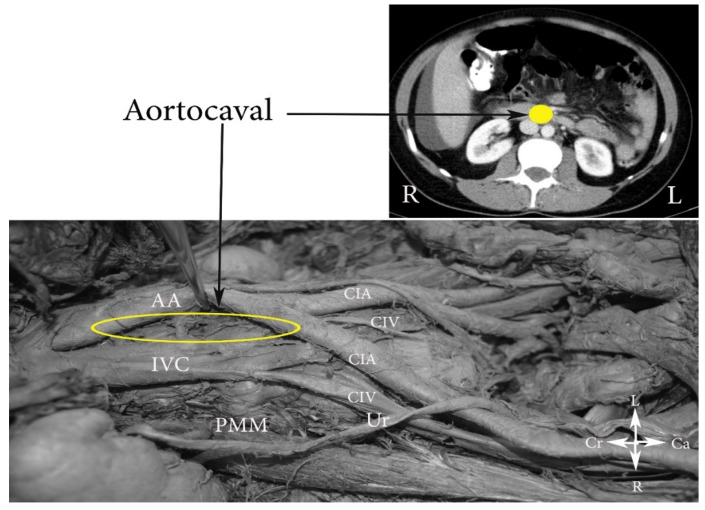
Aortocaval lumbar lymph nodes (embalmed cadaver). CIA—common iliac artery, CIV—common iliac vein, IVC—inferior vena cava, AA—abdominal aorta, PMM—psoas major muscle, Ur—ureter, Cr—cranial, Ca—caudal, L—left, R—right.

**Figure 5 jcm-11-00953-f005:**
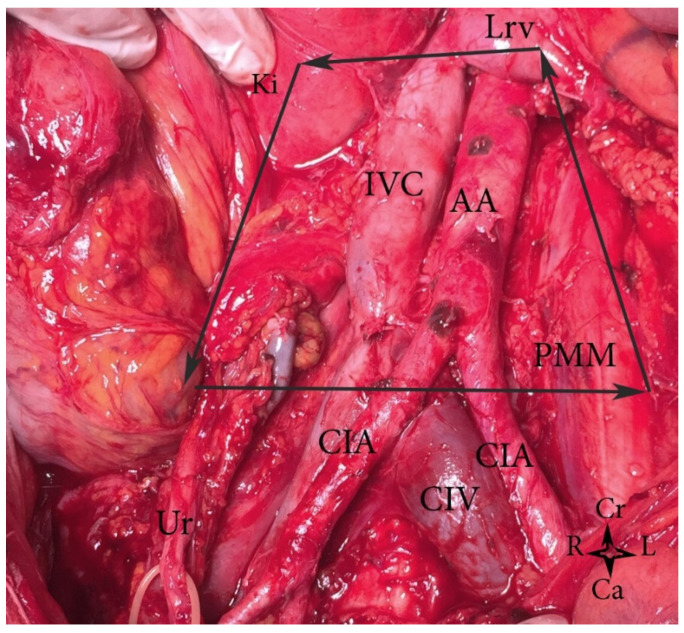
Boundaries of PALND (open surgery). AA—abdominal aorta, IVC—inferior vena cava, CIA—common iliac artery, CIV—common iliac vein, Ur—ureter, Ki—kidney, Lrv—left renal vein, PMM—psoas major muscle, Cr—cranial, Ca—caudal, L—left, R—right.

**Figure 6 jcm-11-00953-f006:**
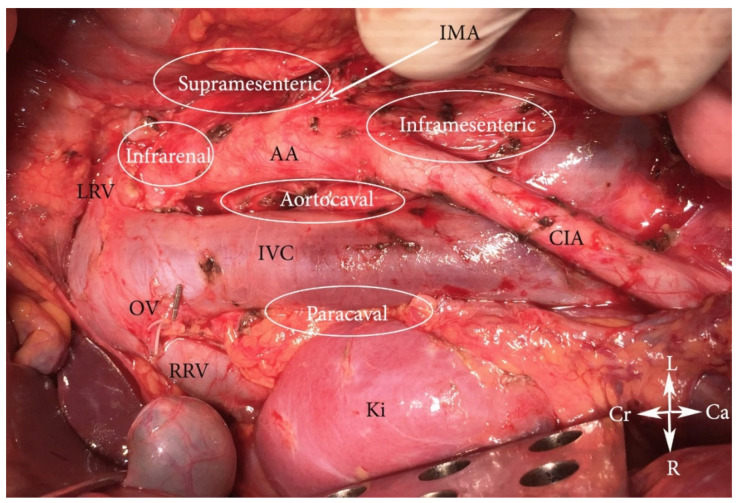
Regions of PALND (open surgery). CIA—common iliac artery, IVC—inferior vena cava, AA—abdominal aorta, RRV—right renal vein; LRV—left renal vein, OV—ligated right ovarian vein at its insertion to the IVC; IMA—inferior mesenteric artery, Ki—kidney, Cr—cranial, Ca—caudal, L—left, R—right.

**Figure 7 jcm-11-00953-f007:**
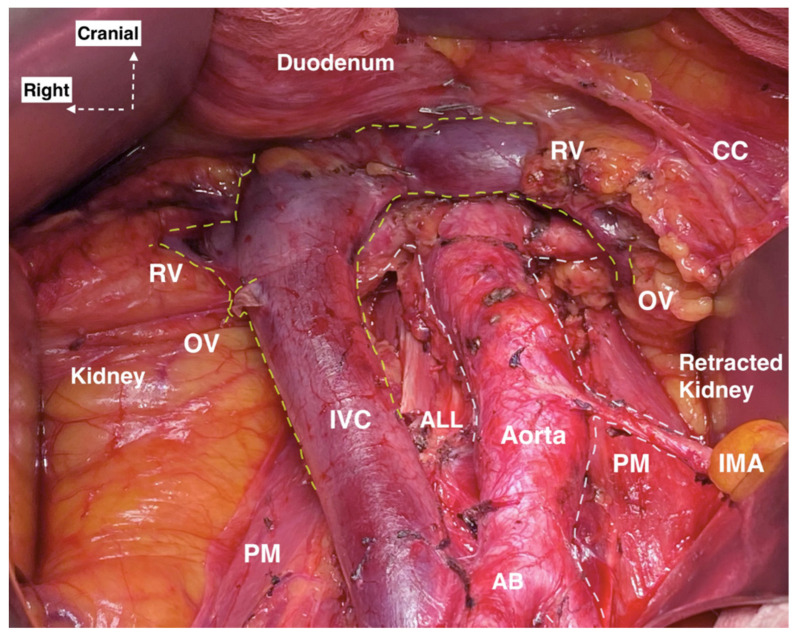
Right and left renal artery lying below the right and left renal vein (open surgery performed by doctor Ilker Selcuk). RV—renal vein, CC—chylous channel, OV—ovarian vein, IVC—inferior vena cava, IMA—inferior mesenteric artery, PM—psoas major muscle, ALL—anterior longitudinal ligament, AB—aortic bifurcation.

**Figure 8 jcm-11-00953-f008:**
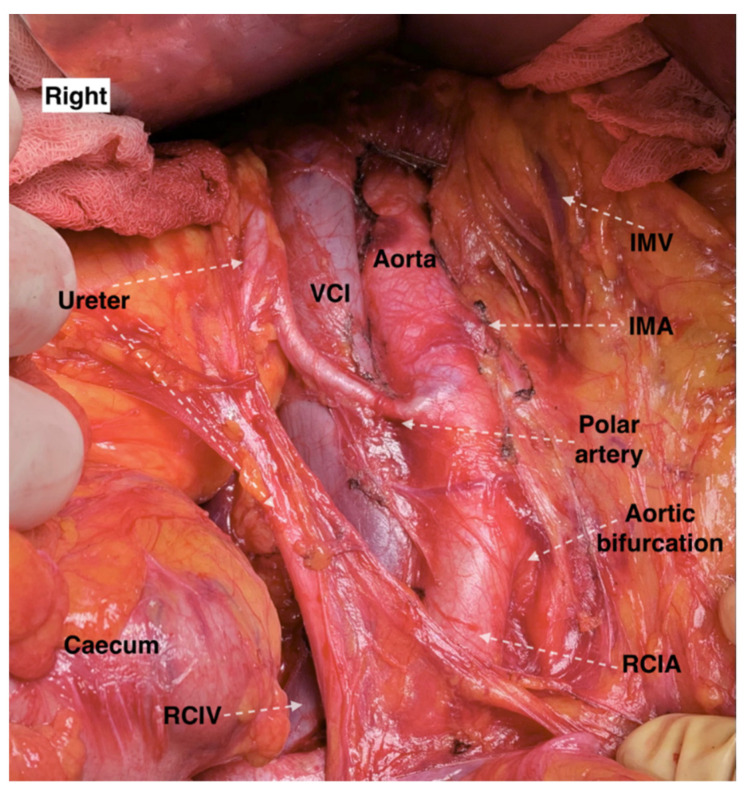
Additional right aortic upper polar renal artery with precaval course, which arises from the anterolateral aspect of the abdominal aorta (an open surgery performed by doctor Ilker Selcuk). IMV—inferior mesenteric vein, IMA—inferior mesenteric artery, RCIA—right common iliac artery, RCIV—right common iliac vein, VCI—vena cava inferior.

**Figure 9 jcm-11-00953-f009:**
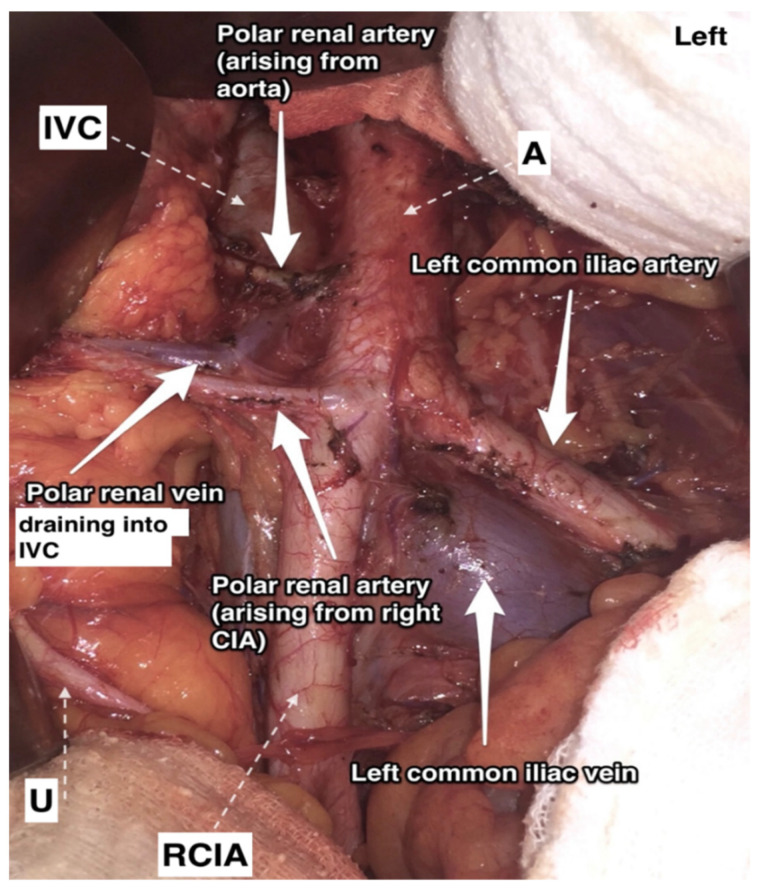
Additional right aortic upper polar renal artery and accessory right aortic lower polar artery arising from the abdominal aorta and the right common iliac artery, respectively. The arteries have a precaval course (open surgery performed by doctor Ilker Selcuk). IVC—inferior vena cava, A—abdominal aorta, U—right ureter, RCIA—right common iliac artery.

**Figure 10 jcm-11-00953-f010:**
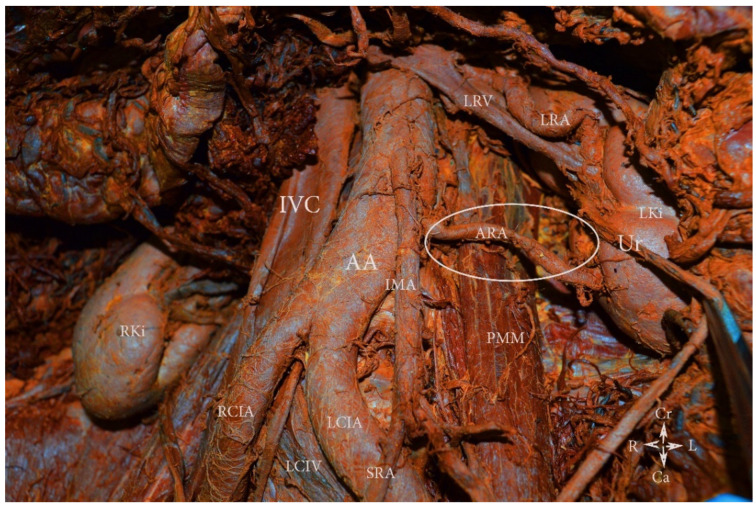
Additional left aortic lower polar renal artery arising from the lateral aspect of the abdominal aorta (embalmed cadaver). The artery originates from the abdominal aorta just below the origin of the inferior mesenteric artery. AA—abdominal aorta, IVC—inferior vena cava, RCIA—right common iliac artery, LCIA—left common iliac artery, LCIV—left common iliac vein, LRV—left renal vein, LRA—left renal artery, ARA—additional left aortic lower polar renal artery, Ur—ureter, LKi—left kidney, RKi—right kidney, PMM—psoas major muscle, IMA—inferior mesenteric artery, SRA—superior rectal artery, Cr—cranial, Ca—caudal, L—left, R—right.

**Figure 11 jcm-11-00953-f011:**
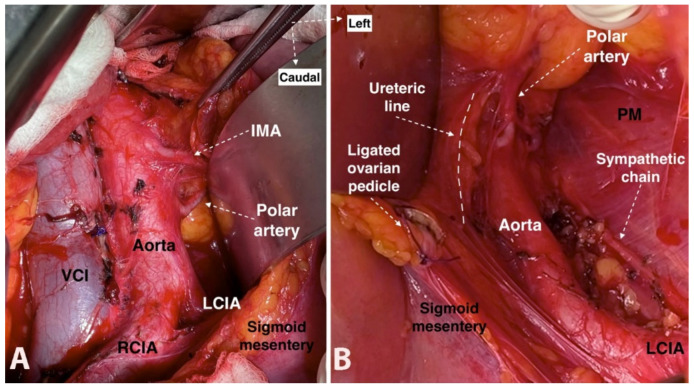
Additional left aortic lower polar renal artery lying towards the left kidney, which arises caudal to the inferior mesenteric artery from the lateral aspect of abdominal aorta (open surgery performed by doctor Ilker Selcuk). (**A**)—The additional left aortic lower polar renal artery after lateralization of the sigmoid mesentery. (**B**)—The additional left aortic lower polar renal artery after medial mobilization of the sigmoid mesentery. IMA—inferior mesenteric artery, VCI—vena cava inferior, LCIA—left common iliac artery, RCIA—right common iliac artery, PM—psoas major muscle.

**Figure 12 jcm-11-00953-f012:**
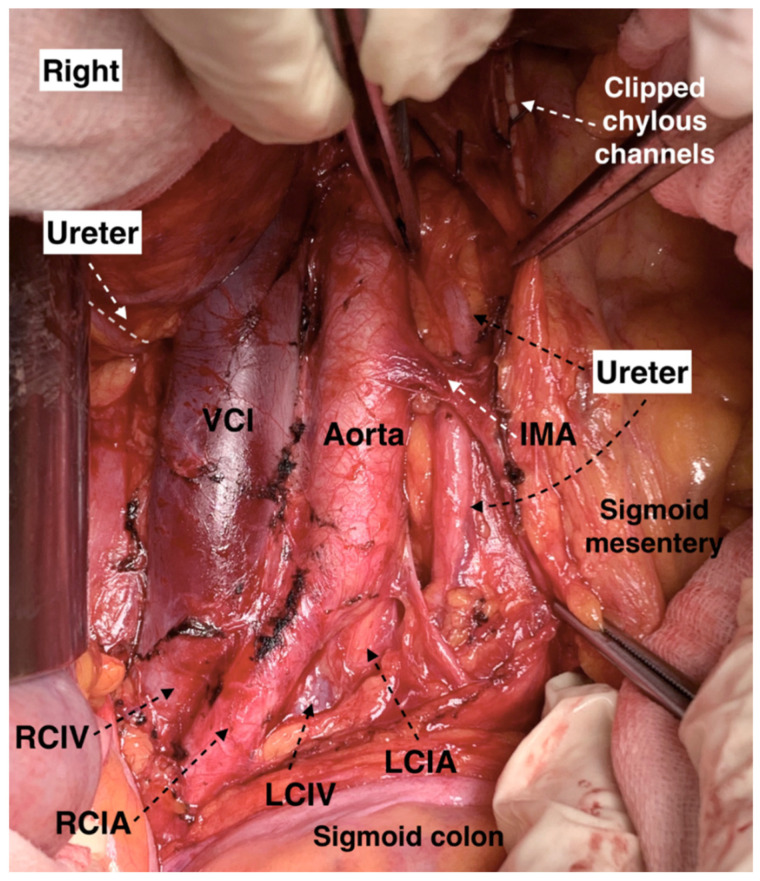
Left ureter at the lateral aspect of the abdominal aorta, which is identified under the mesentery of the sigmoid colon (open surgery performed by doctor Ilker Selcuk). IMA—inferior mesenteric artery, LCIA—left common iliac artery, LCIV—left common iliac vein, RCIV—right common iliac vein, RCIA—right common iliac vein, VCI—vena cava inferior.

**Figure 13 jcm-11-00953-f013:**
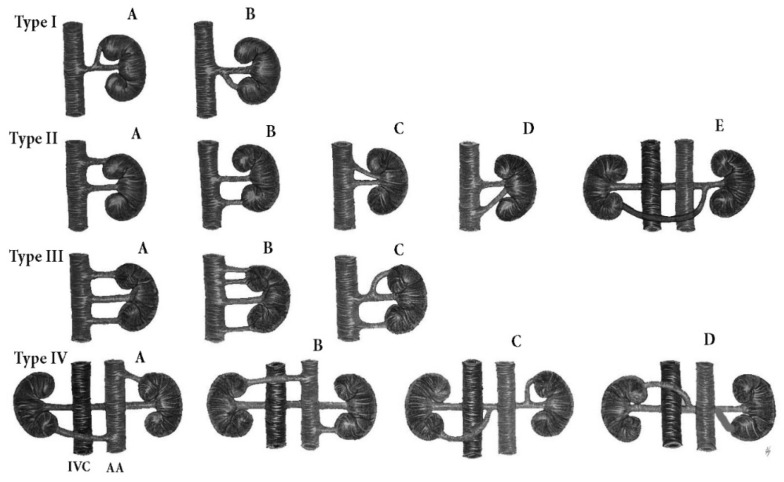
Additional renal arteries classification. Type I—additional renal arteries arising from the main renal artery. IA—additional upper polar renal artery, IB—additional lower polar renal artery. Type II—additional renal arteries arising from the abdominal aorta or renal vein, IIA—additional aortic upper polar renal artery, IIB—additional aortic lower polar renal artery, IIC—additional aortic upper hilar renal artery, IID—additional aortic lower hilar renal artery, IIE—additional lower polar renal artery arising from the left renal artery and entering the right kidney. The artery has a preaortic and precaval course. Type III—first and second additional renal arteries. Only polar arteries are shown, but the arteries may enter the hilar or polar part of the kidney. IIIA—first additional arteries arising from the abdominal aorta, IIIB—second additional renal arteries arising from the abdominal aorta, IIIC—first additional renal arteries arising from the abdominal aorta and renal artery. Type IV—bilateral additional renal arteries. Same as Type III, only polar arteries are shown, but the presence of both hilar and polar arteries is possible. IVA, B—bilateral additional renal arteries arising from the abdominal aorta. IVC, D—bilateral additional renal arteries arising from the renal artery. IVC—inferior vena cava, AA—abdominal aorta.

**Figure 14 jcm-11-00953-f014:**
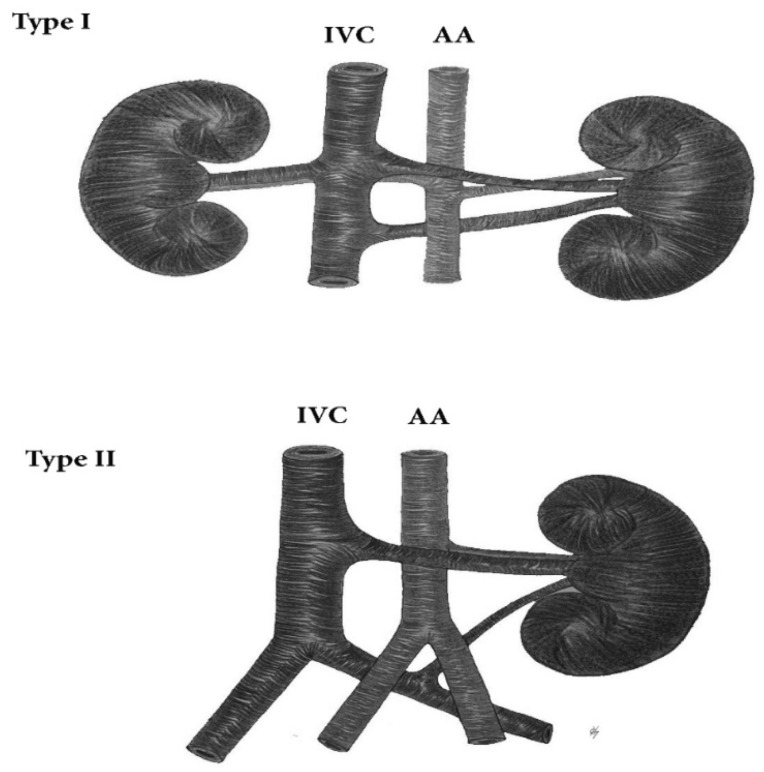
Circumaortic left renal vein. Type I—the additional renal vein has a retroaortic course and drains in the IVC. Type II—the additional renal vein passes behind the left common iliac artery and drains in the left common iliac vein. IVC—inferior vena cava, AA—abdominal aorta.

**Figure 15 jcm-11-00953-f015:**
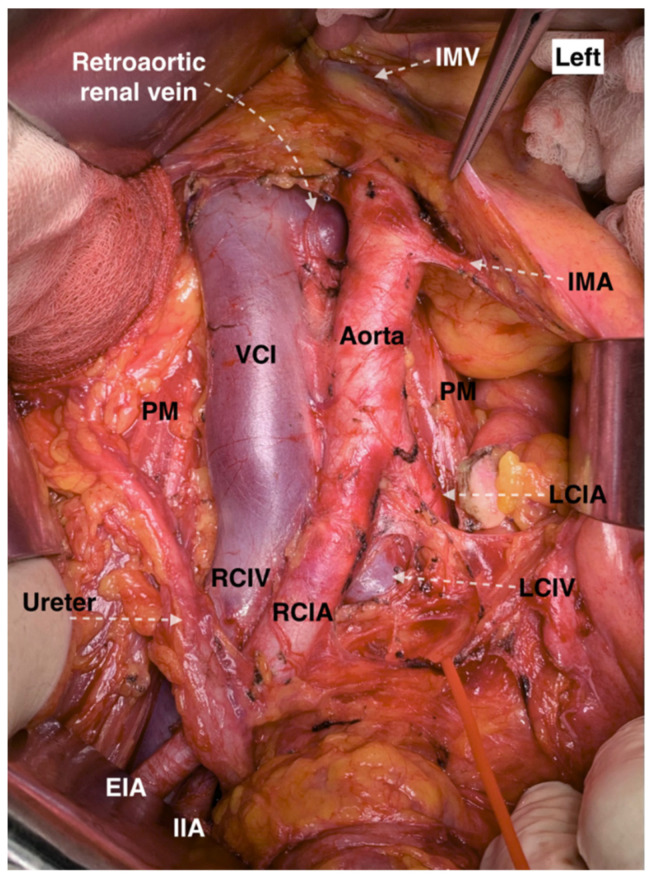
Retroaortic left renal vein at the aortocaval region (open surgery performed by Ilker Selcuk). VCI—vena cava inferior, IMA—inferior mesenteric artery, PM—psoas major muscle, LCIA—left common iliac artery, LCIV—left common iliac vein, RCIA—right common iliac artery RCIV—right common iliac vein, EIA—external iliac artery, IIA—internal iliac artery, IMV—inferior mesenteric vein.

**Figure 16 jcm-11-00953-f016:**
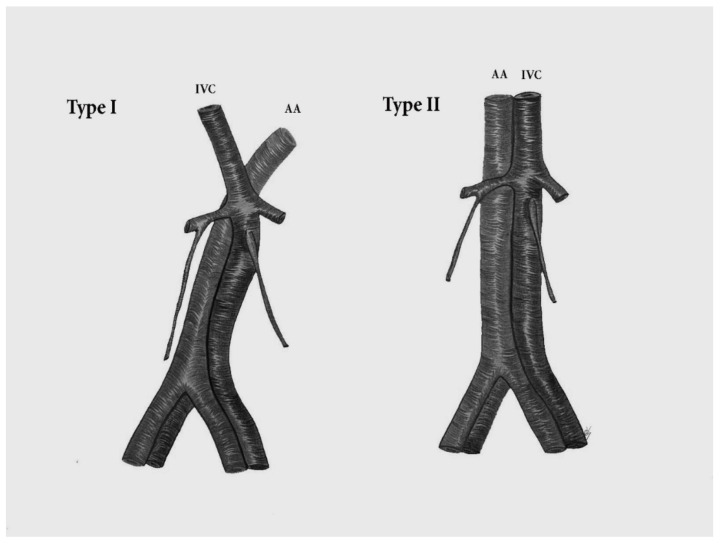
Left sided inferior vena cava. Type I—the IVC passes anterior to the level of the LRV to form a normal right-sided IVC. Type II—the IVC does not cross the aorta and continues its passage upward. The left sided IVC is a mirror image of normal anatomy—the right ovarian vein drains into the RRV and the left ovarian vein drains into the IVC. IVC—inferior vena cava, AA—abdominal aorta.

**Figure 17 jcm-11-00953-f017:**
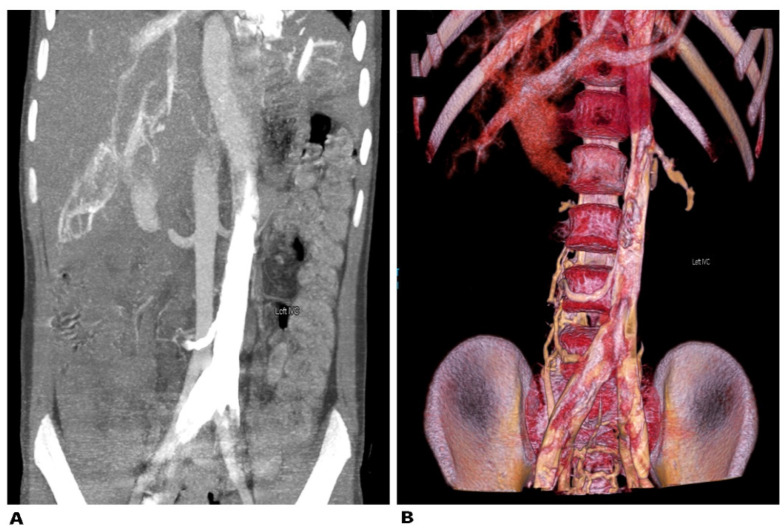
Left sided inferior vena cava (Type II). (**A**)—abdominal CT in the coronal plane. The contrast-enhanced left vena cava is positioned on the left side. (**B**)—3D volume-rendered reconstruction in coronal view.

**Figure 18 jcm-11-00953-f018:**
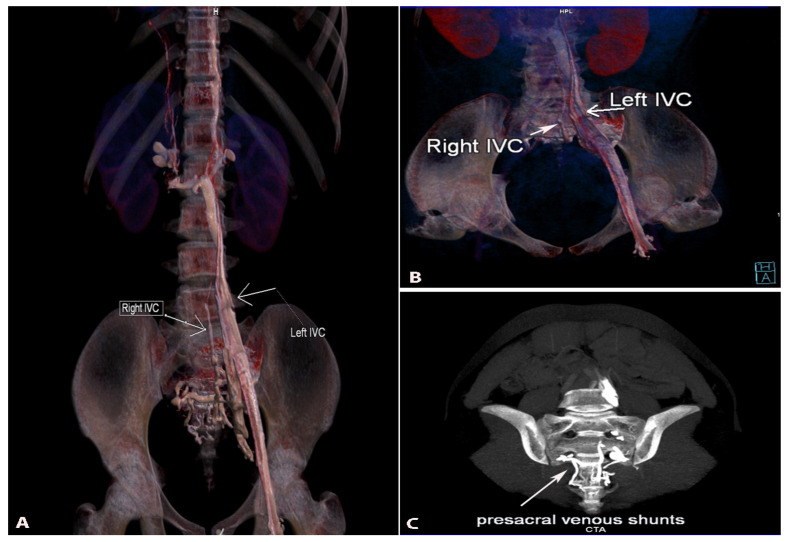
Double inferior vena cava with regressed right inferior vena cava. (**A**)—3D volume rendered reconstruction in coronal view. Right-sided IVC is regressed with multiple presacral venous communications between both caval systems. (**B**)—3D volume rendered reconstruction in coronal oblique view. Left dominant inferior vena cava is annotated with an arrow. The right regressed inferior vena cava is the vessel just in front of the spine. (**C**)—maximum intensity projection—oblique view parallel to coronal sacral plane. Multiple venous shunts are seen occupying the presacral region.

**Figure 19 jcm-11-00953-f019:**
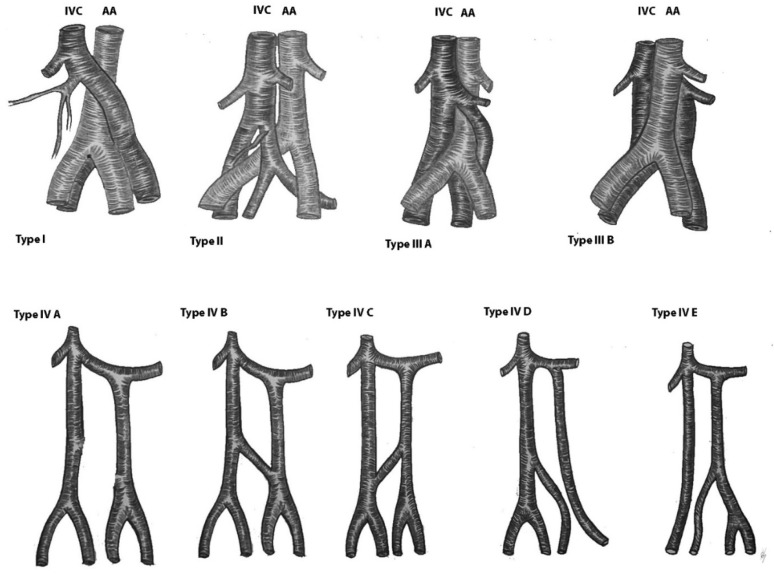
Variations of the duplicated inferior vena cava. Type I—Left ICV with regressed right IVC. Type II—right-sided DIVC. The ovarian vein drains into the ventral IVC. Type III. BDIVC was located anterior or posterior to the AA. The left IVC drains into the right IVC. Type IIIA—the left IVC passes in front of the AA at the level of LRV and drains in the right IVC. Type IIIB—The left IVC passes behind the AA at the level of LRV and drain in the right IVC. Type IV—BDIVC where the left vena cava enters the LRV. Type IVA—absence of communication between left and right IVC. Type IVB—interiliac communication between DBIVC from the left common iliac vein. Type IVC—interiliac communication between DBIVC from the right common iliac vein. Type IVD—interiliac communication between DBIVC from the left internal iliac vein. Type IVE—interiliac communication between DBIVC from the right internal iliac vein.

**Figure 20 jcm-11-00953-f020:**
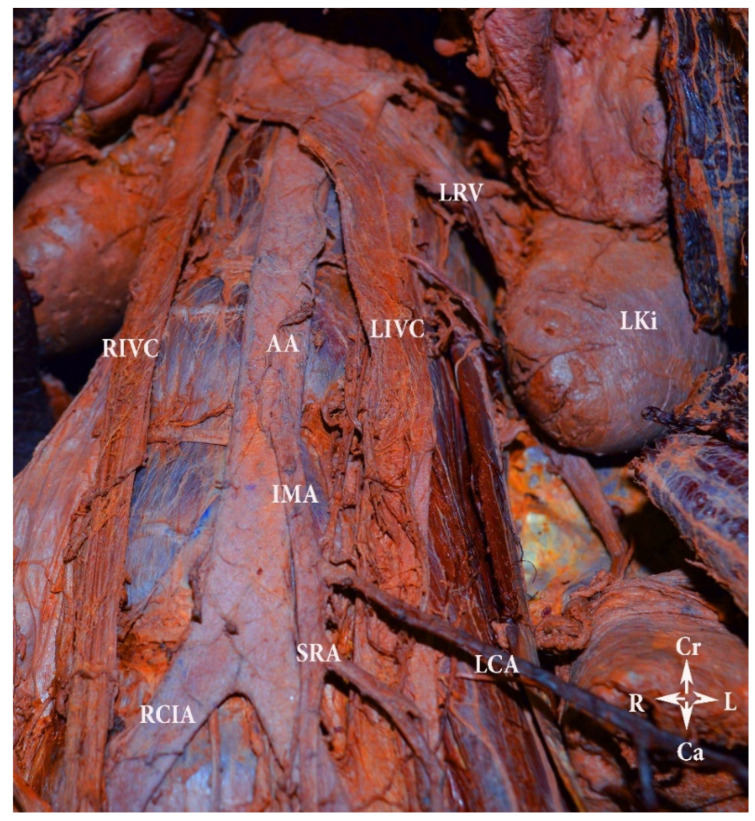
Bilateral duplication of the inferior vena cava with no communication between veins (embalmed cadaver). Type IVA according to our classification. LIVC—left inferior vena cava, RIVC—right inferior vena cava, AA—abdominal aorta, LCA—left colic artery, SRA—superior rectal artery, IMA—inferior mesenteric artery, RCIA—right common iliac artery, LRV—left renal vein, LKi—left kidney, Cr—cranial, Ca—caudal, L—left, R—right.

**Figure 21 jcm-11-00953-f021:**
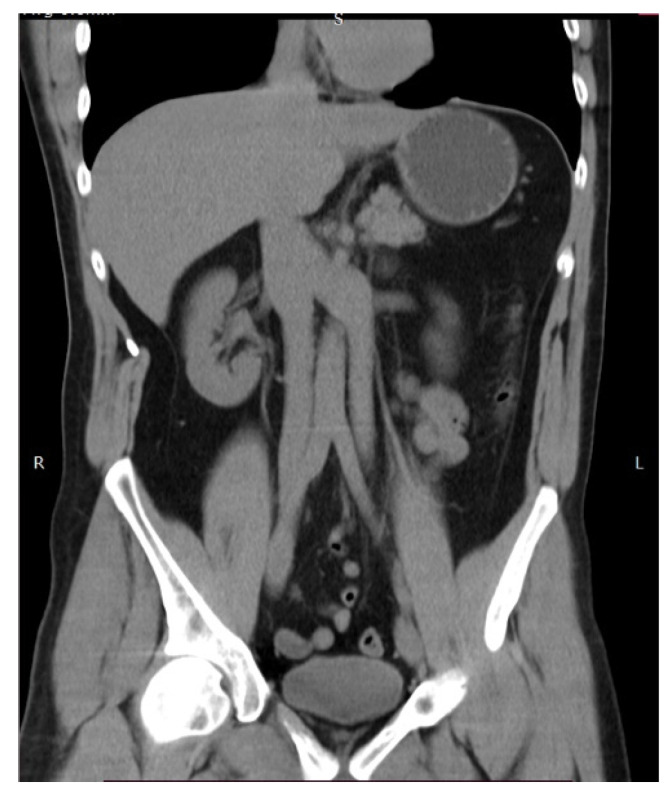
Bilateral double inferior vena cava (Type IIIA). Abdominal CT in the coronal plane.

**Table 1 jcm-11-00953-t001:** Vessel variations in the paraaortic regions. AA—abdominal aorta, IVC—inferior vena cava, LOV—left ovarian vein, LLVs—left lumbar veins, ROV—right ovarian vein, RRV—right renal vein, IMA—inferior mesenteric artery, LVs –lumbar veins, LRV—left renal vein.

Vessel Variations	Incidence	Risk of Injury in the Paraaortic Regions	Reported Incidence of Injury
**Renal arteries**			
*Additional renal arteries*	10–50%	Infra/supramesenteric, aortocaval regions	Unknown
*Accessory renal arteries*	Unknown	All regions	Unknown
*Precaval right renal artery*	0.8–5%	Infra/supramesenteric, aortocaval	Unknown
**Renal veins**			
*Additional renal veins*	Left—1.3–3.2% Right—20–23%	All regions	Unknown
*Circumaortic left renal vein*	3.5–7%	Infra/supramesenteric, aortocaval regions	50%
*Retroaortic left renal vein*	1.84–6.6%	Infra/supramesenteric, aortocaval regions	19%
*Retropelvic tributary of the left renal vein*	30.0–46.4%	Infra/supramesenteric region	Unknown
**Abdominal aorta**			
*Double AA*	Only few cases described	All regions	Unknown
**Inferior vena cava**			
*Left sided IVC*	0.2–0.5%	Infra/supramesenteric region	Unknown
*Left IVC with regressed right IVC*	Less than 10 cases described	Infra/supramesenteric, caudal aortocaval regions	Unknown
*Right sided duplication of IVC*	Approximately 10 cases described	Aortocaval, paracaval	Unknown
*Bilateral duplication of IVC*	0.2–3%	All regions	Unknown
*Marsupial IVC*	Approximately 20 cases described	Aortocaval, paracaval	Unknown
**Ovarian vessels**			
*Ovarian arteries*	Unknown	All regions	Unknown
**Ovarian veins**			
*LOV draining into the LLVs*	2.2%	Infra/supramesenteric	Unknown
*ROV draining into the RRV*	8.8–9.9%	Paracaval	Unknown
**IMA**			
*IMA arising from SMA*	0.1%	Aortocaval, Infra/supramesenteric regions	Unknown
**LVs**			
LVs draining into the LRV	17.4–49%	Aortocaval, Infra/supramesenteric	Unknown

## Data Availability

Authors declare that all related data are available concerning researchers by the corresponding author’s email.

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
