# Peer review of "Paraaortic Lymphadenectomy in Gynecologic Oncology—Significance of Vessels Variations"

_jcm, 2022, doi:10.3390/jcm11040953_

Round 1
Reviewer 1 Report
Dear Authors,
Congratulations, your anatomical review took me back to my Grey's anatomy days (which I loved). Your detailed examination of the para-aortic anatomy is a true "tour-de-force".
Some minor English editing is required especially in regards to verb tense, words like "this vs these", use of "less" instead of "fewer", etc.
My major criticism of this manuscript relates to defining it's objective. I believe defining the extent of a totally complete P/A dissection to set the margins for the anatomical discussion is perfectly reasonable. I do think a single statement early in the manuscript that says "the extent of the dissection required for an individual case, should be determined by a gynecologic oncologist who is totally versed in the disease, it's pathologic grade, and the effectiveness of all available non-surgical therapies". Rarely, possibly never, is a para-aortic dissection of the type described required in gyn oncology. The surgery must have benefit. Gynecologic cancers are among the most responsive of all abdominal solid tumors to chemotherapy and radiation. Therefore, in the presence of regional spread, especially bulky sidewall disease, for example in cervical cancer, the delay in instituting definitive therapy (radiation/platinum), or the additional risk of radiation complications after an extensive para-aortic dissection to find microscopic disease when no enlarged nodes are seen on CT, will generally not warrant the risks and treatment delay. I have cared for many patients over my career, whose primary treatment was designed by a physician who did not understand gyn cancers, and a very skilled surgeon who did some extensive procedure that was unnecessary and missed the key therapeutic steps or delayed definitive therapy. Once bulk nodal disease is present, talking about skip micro-metastases in patients who clearly need systemic therapy as a justification for more extensive surgery is misguided. My point being, since you have directed this manuscript to gynecologic cancers, your wonderful anatomic tour should have a clear statement relating to the importance of balancing risk with benefit for particular gyn cancers, which are in most peoples view gradually becoming more non-surgical. In addition the morbid obesity which is frequent in patients with endometrial cancers and the proliferation of robotics in today's surgery adds further complexity to these decisions specific to gynecologic cancers.
Author Response
Reviewer 1.
Congratulations, your anatomical review took me back to my Grey's anatomy days (which I loved). Your detailed examination of the para-aortic anatomy is a true "tour-de-force".
- Some minor English editing is required especially in regards to verb tense, words like "this vs these", use of "less" instead of "fewer", etc.
Author’s Reply:
A native English speaker corrected errors in grammar, punctuation, word choice, and sentence construction to improve the flow of ideas expressed in the article.
- My major criticism of this manuscript relates to defining it's objective. I believe defining the extent of a totally complete P/A dissection to set the margins for the anatomical discussion is perfectly reasonable. I do think a single statement early in the manuscript that says "the extent of the dissection required for an individual case, should be determined by a gynecologic oncologist who is totally versed in the disease, it's pathologic grade, and the effectiveness of all available non-surgical therapies". Rarely, possibly never, is a para-aortic dissection of the type described required in gyn oncology. The surgery must have benefit. Gynecologic cancers are among the most responsive of all abdominal solid tumors to chemotherapy and radiation. Therefore, in the presence of regional spread, especially bulky sidewall disease, for example in cervical cancer, the delay in instituting definitive therapy (radiation/platinum), or the additional risk of radiation complications after an extensive para-aortic dissection to find microscopic disease when no enlarged nodes are seen on CT, will generally not warrant the risks and treatment delay. I have cared for many patients over my career, whose primary treatment was designed by a physician who did not understand gyn cancers, and a very skilled surgeon who did some extensive procedure that was unnecessary and missed the key therapeutic steps or delayed definitive therapy. Once bulk nodal disease is present, talking about skip micro-metastases in patients who clearly need systemic therapy as a justification for more extensive surgery is misguided. My point being, since you have directed this manuscript to gynecologic cancers, your wonderful anatomic tour should have a clear statement relating to the importance of balancing risk with benefit for particular gyn cancers, which are in most peoples view gradually becoming more non-surgical. In addition the morbid obesity which is frequent in patients with endometrial cancers and the proliferation of robotics in today's surgery adds further complexity to these decisions specific to gynecologic cancers.
Author’s Reply:
We totally agree with the reviewer. Multidisciplinary team should carefully discuss the extent of radicality. Nowadays the extreme radicality is not often perfomred and peri-operative management provides information about the possible morbidity of the patient – age, albumin levels, BMI, etc. Our opinion is the same as the reviewer. It has been incorporated in the text. We chose to put this paragraph just after the complications of PALND. When readers get familiar with the high rate and different types of complications, they will understand why surgeons try to avoid extent radicality.
The next text was inserted:
It should be stressed that the extent of the PALND required for an individual case, should be determined by an oncogynecologists who is totally experienced and familiar with the disease (individual peri-operative management of patients and histological type, pathologic grade, molecular characteristics of the tumors) and the effectiveness of all available non-surgical therapies. Therefore, of utmost importance is whether the risks of a systematic PALND outweigh the benefits. Extent radicality is not always the key to better survival and may increase mortality. In order to decrease morbidity, and to improve oncologic care in women with ovarian cancer, guidelines for peri-operative management has been recently introduced. [26]. Currently, studies which showed that less radicality is associated with similar outcomes compared to radical surgery, were widely accepted [25, 27]. The lymphadenectomy in ovarian neoplasms (LION) trial proved that patients with normal lymph nodes, who underwent PALND were not associated with increased overall or progression-free survival than patients in the no lymphadenectomy group. Moreover, more postoperative complications were observed in patients with the lymphadenectomy group [25]. Additionally, another study reported that completion of radical hysterectomy is not associated with an increased survival in women with intraoperatively detected lymph node involvement, regardless of the size of the tumour or histology [27].
In conclusion, multidisciplinary team management and appropriate surgical selection criteria are needed in order to improve patient outcome.
Two references were inserted: 26, 27.
We are grateful for your valuable time and effort in reviewing our manuscript.
Based on your useful and scientific comments, we believe our manuscript has been improved to a higher level.
Reviewer 2 Report
- For preoperative diagnostic of vessels abnormalities or variants, what is the rate, if known, of imaging false negatives? (my apologies if it is already in the text). This would help to stress the importance of a good collaboration between surgeons and radiologists.
- Intraoperative ultrasound with pulsed and color Doppler has already been used in different retroperitoneal malignancies to determine their resectability during surgical exploration. Could it be of additional value to find and better define vascular abnormalities?
Author Response
- For preoperative diagnostic of vessels abnormalities or variants, what is the rate, if known, of imaging false negatives? (my apologies if it is already in the text). This would help to stress the importance of a good collaboration between surgeons and radiologists.
Author’s Reply:
We totally agree with the reviewer that the rate of false negative results on imaging modalities is an important topic question! Unfortunately, there are a few studies (most of them are case studies), which clearly described the rate of false positive and false negative results. However, we manage to find a few articles. Generally, as AA and IVC are large vessels false negative results are rare, whereas in renal vessels the rate of false negative results is higher. For the IMA there is not such a study, whereas other vessels ( lumbar, ovarian) are very small vessels and radiologists do not pay much attention for such a variations ( it is hard and injury is not related to catastrophic complications). In our opinion the most important vessels, which are associated with false-negative results are IVC and renal vessels. That’s why they are discussed in the paragraph below.
The next text was incorporated:
It should be stressed to surgeons that there is possibility of false-negative results on imaging. False- negative results are less likely to be observed in large vessels such as the IVC and the AA. However, there are case studies reporting a preoperative imaging failure to recognized bilateral duplication of the IVC [94, 123]. Moreover as it was mentioned above variant IVC could be misdiagnosed with different retroperitoneal pathologies [95-97]. In contrast, the false-negative results are higher for small vessels variations, especially if the vessels are occluded and nonfunctional. In our opinion, it is dangerous for the surgeon if the renal vessels variations are not recognized preoperatively in CT or other imaging modality. Hassan et al. examined 56 patients for renal vessels variation by a multi detector computed tomography angiography. The protocol of the study acquired CT data in all phases (arterial, venous and delayed). Authors observed 5.4% of false negative results in renal veins variations, whereas for arterial variations the results were 1.8%, respectively [124].
Two new references was inserted: 123, 124.
- Intraoperative ultrasound with pulsed and color Doppler has already been used in different retroperitoneal malignancies to determine their resectability during surgical exploration. Could it be of additional value to find and better define vascular abnormalities?
Author’s Reply:
Interesting question! Intraoperative ultrasound has been widely accepted in all surgical specialties (brain, spine, breast, pancreas and liver, gynecologic surgery). During hepatic resections, intraoperative ultrasound helps surgeons visualize complex anatomical structures and variations. In gynecologic oncology, it is mainly used for intraoperative sonographic guidance for intracavitary brachytherapy of cervical cancer, assessment of myometrial invasion or for liver metastases.
We cannot find an article dealing with intraoperative ultrasound (IOUS) and anatomical variations in Gynecologic oncology, but it definitely should be mentioned that it is possible and it may be useful in the future (since it is used in other surgeries for anatomical variations – brain, liver). False-negative value of almost 6 % for renal anomalies is a high number. Therefore, in our opinion IOUS could be used especially at that region.
The next text was inserted:
As the incidence of false-negative results is not low, intraoperative ultrasound, (IOUS) may be of an additional value to find and better define vascular abnormalities, especially in the renal region. Currently, IOUS has been widely used in gynecologic oncology – assessment of myometrial invasion, avoiding unnecessary PALND in women with endometrial cancer, detection and localization of recurrent gynecological cancer, etc [125-127]. Although there is no study in gynecologic oncology describing IOUS and anatomical variations, in general surgery there are many examples [128, 129]. Therefore, in our opinion IOUS could be a useful tool for detecting suspected anatomical variations in gynecologic oncology.
Five new references were incorporated : 125-129
We are grateful for your valuable time and effort in reviewing our manuscript.
Based on your useful and scientific comments, we believe our manuscript has been improved to a higher level.